# Silicon/Hard Carbon Composites Synthesized from Phenolic Resin as Anode Materials for Lithium-Ion Batteries

**DOI:** 10.3390/nano15060455

**Published:** 2025-03-17

**Authors:** Yu-Hsuan Li, Sompalli Kishore Babu, Duncan H. Gregory, Soorathep Kheawhom, Jeng-Kuei Chang, Wei-Ren Liu

**Affiliations:** 1Department of Chemical Engineering, R&D Center for Membrane Technology, Chung Yuan Christian University, 200 Chung Pei Road, Chungli District, Taoyuan City 32023, Taiwan; levatw@gmail.com (Y.-H.L.); kishoresompalli96@gmail.com (S.K.B.); 2WestCHEM, School of Chemistry, University of Glasgow, Glasgow G12 8QQ, UK; duncan.gregory@glasgow.ac.uk; 3Department of Chemical Engineering, Faculty of Engineering, Chulalongkorn University, Bangkok 10330, Thailand; soorathep.k@chula.ac.th; 4Center of Excellence on Advanced Materials for Energy Storage, Chulalongkorn University, Bangkok 10330, Thailand; 5Department of Materials Science and Engineering, National Yang Ming Chiao Tung University, 1001 University Road, Hsinchu 30010, Taiwan; jkchang@nycu.edu.tw; 6Institute of Materials Science and Engineering, National Central University, 300 Jhong-Da Road, Taoyuan 32001, Taiwan; 7Hierarchical Green-Energy Materials (Hi-GEM) Research Center, National Cheng Kung University, 1 University Road, Tainan 70101, Taiwan

**Keywords:** hard carbon, silicon, pitch coating, anode, Li-ion batteries

## Abstract

Silicon could revolutionize the performance of lithium-ion batteries (LIBs) due to its formidable theoretical gravimetric capacity, approximately ten times that of graphite. However, huge volume expansion during charge/discharge processes and poor electronic conductivity inhibited its commercialization. To address the problems, new carbon-silicon core-shell microparticles have emerged for prospective anodes in LIBs. In this study, we develop a core-shell structure by using hard carbon derived from phenolic resin as the core and nano silicon/pitch coating as the shell to the resulting HC@Si-P composite anode. A composition-optimized 20 wt.% pitch coated-Si/HC composite anode delivers superior cycling stability over 200 cycles under 1 A/g current density, showing a 398 mAh/g capacity. At 5.0 A/g current density during charge and discharge processes, the reversible capacity reaches 215 mAh/g. Upon reducing the current density to 0.1 A/g, the capacity remains high at 537 mAh/g. Impedance testing shows that after pitch coating, the RSEI impedance decreases and the diffusion coefficient of HC@Si-P increases. Moreover, the facile and scalable preparation technique is encouraging for the potential practical application of silicon-based anode materials of this type in the upcoming generation of LIBs.

## 1. Introduction

Lithium-ion batteries (LIBs) are extensively utilized across various applications, including consumer electronics, grid-scale energy storage, and electric vehicles. Their widespread adoption is attributed to several critical advantages, such as high energy density, extended cycle life, and a low self-discharge rate. These attributes make LIBs a preferred choice in energy storage and mobility solutions [1,2]. Despite their advancements, current lithium-ion battery technology faces significant limitations that affect their practical use. This limitation means electric vehicles often require more frequent charging and their energy storage systems cannot hold as much power as petrol or diesel-powered fuel tanks. Consequently, these factors impact the convenience and practicality of long-distance travel or heavy-duty use [3,4]. The energy density of a battery is primarily determined by the specific capacities of the materials used in its anode and cathode. As a result, there is a growing demand for energy storage materials with higher capacities to address the expanding energy requirements of modern society. As technology advances and reliance on portable and renewable energy sources expands, developing more efficient, high-capacity materials has become critical to ensuring batteries can store sufficient energy while maintaining performance and longevity [5,6]. Compared to graphite, hard carbon (HC) provides better rate performance during cycling because its structure allows for relatively unrestricted 3D ion transfer pathways [7]. HC and graphite are carbon-based materials, but their properties and structures differ due to distinct atomic arrangements and bonding. Unlike the crystalline structure of graphite, HC has an amorphous structure with larger d-spacing. Hard carbon is a promising anode material for lithium-ion batteries (LIBs), offering several advantages, including high energy storage capacity, cost-effectiveness, stable performance across a range of charge/discharge, excellent thermal stability, and long cycle life. These properties make hard carbon a strong candidate for LIB applications, as it ensures efficient energy retention, withstands operational stresses over time, and offers a cost-effective solution for large-scale production. Additionally, its thermal stability helps maintain battery safety [8]. HC anodes also exhibit a low potential plateau with a significant capacity at ≤0.2 V (vs. Li/Li^+^), enabling them to deliver high power density in LIBs [9,10,11,12]. Although HC offers advantages over graphite, the performance of LIB anodes can only be enhanced to a certain extent. They must go beyond carbon to significantly improve reversible capacity, rate capability, and cycle life, which is crucial for the widespread use of next-generation anode materials. Silicon [13,14,15,16,17,18,19,20,21,22,23,24,25,26,27], tin [28,29], and selected metal oxides [30,31,32] are among the leading contenders for alternative anode materials. Although these materials offer significantly higher theoretical capacities than carbon, using them as conversion or alloying electrodes often presents challenges such as reduced stability and substantial volume expansion during the lithiation and de-lithiation processes. This dramatic volume expansion can cause mechanical stress, leading to electrode degradation and a shorter battery lifespan. Moreover, the instability of these materials during repeated cycling leads to performance degradation, thereby restricting their practical applicability in energy storage devices [33]. Considering silicon, which has an exceptionally high theoretical gravimetric capacity (4008 mAh/g for Li_22_Si_5_) [34], several approaches have been proposed to combat capacity loss during (de)alloying, which include carbon coating [35,36,37] and the design of porous structures at the micro and nanoscales [8,38,39].

In this vein, Hu. et al. utilized a surface oxidation strategy to improve the electrochemical performance of Si nanoparticles by forming a SiOx layer on their surface. This layer facilitated the subsequent coating of the nanoparticles with N-doped carbon through hydrogen bonding [40]. The nitrogen-doped carbon coating enhanced the material’s ability to conduct electricity while reinforcing its overall structure. The SiOx layer helps ensure an even, uniform resin coating, which can later be transformed into a nitrogen-doped carbon shell. Correspondingly, the surface oxidation strategy resulted in an anode with sustained performance, a high reversible capacity of 739 mA h g^−1^ after 500 cycles, and a more than 99% coulombic efficiency. Lu. et al. investigated a carbon-coated silicon core-shell nanocomposite that was prepared using two steps, including a self-assembly process and a subsequent sintering treatment. The mixture of silicon nanoparticles and phenolic resin had a weight ratio of 4:1, meaning there were four silicon nanoparticles for every part of the phenolic resin. After carbonization, the material contained a high silicon content, reaching 86% by weight. This composition-optimized electrode demonstrated a reversible capacity of 1108 mAh/g at 0.3 C over 50 cycles. It maintained nearly 1000 mAh/g over 200 cycles, achieving a coulombic efficiency exceeding 99.6% [41]. Huang et al. utilized a strategy of preparing anodes from “silicon–graphene–carbon” composites for use in LIBs [42]. The composite material is synthesized using a “surface charge self-assembly method,” which integrates nano-silicon and phenolic resin to create a carbon framework interspersed with graphene oxide. The nano-silicon particles have a sheet-like structure, measuring between 30 and 150 nm, and exhibit slight agglomeration. A carbon coating is applied to the surface of nano-silicon particles using a chemical vapor deposition (CVD) technique. The proposed CVD treatment results in a porous structure that significantly improves the cycling performance of the material. The integration of graphene with porous silicon helps to reduce polarization during cycling. A half-cell featuring an anode composed of silicon–graphene–carbon (SGC) achieved a discharge-specific capacity of 1465 mAh/g, retaining 81.3% of this capacity after 100 cycles at a current density of 0.25 C. The battery cell uses sugar-derived graphite (SGC) as the anode to deliver a specific discharge capacity of 550 mAh/g, retaining over 80% of its capacity even after 800 charge-discharge cycles.

Additionally, phenolic resin has been employed to coat silicon within a silicon/carbon (Si/C) nanocomposite by dispersing silicon nanoparticles in a phenolic resin gel, followed by pyrolysis to form a protective carbon layer. This process enhances the structural stability and cycle performance of silicon anodes, which, despite their high capacity, are prone to volume expansion and degradation during cycling [43]. The composition-optimized Si/C nanocomposite demonstrated an initial reversible capacity of 904 mAh/g. It maintained a stable capacity at 75% of this value after 50 cycles at a current density of 100 mA g^−1^. In 2020, a thorough study was conducted on silicon–carbon (Si@C) anodes for lithium-ion batteries. This study focused on how these anodes developed through the pyrolysis of phenolic resins, perform and their potential benefits. Pyrolysis is a process where organic materials are decomposed at high temperatures in the absence of oxygen, which, in this case, transforms phenolic resins into a silicon–carbon composite with potentially improved performance characteristics for battery applications [44].

Our previous research showed that designing various silicon/carbon (Si/C) structures in composite anodes could substantially enhance the performance of the electrodes. This improvement arises because different structural configurations of Si and C in the composite anodes can better accommodate the stresses and strains during charge and discharge cycles, leading to more efficient and stable battery operation [45,46,47,48,49,50]. Herein, we introduce a new type of Si/C composite in which a shell of amorphous, “soft” carbon is assembled around a nano-silicon/hard carbon core. Combining an active composite core and a pitch-derived shell buffers silicon’s volume expansion and enhances the anode’s electronic conductivity. This unique Si/C composite represents a promising strategy for advancing anode designs in lithium-ion batteries.

## 2. Materials and Methods

### 2.1. Material Preparations

Hard carbon (HC) and HC-silicon core-shell composites (HC@Sis) can be synthesized through relatively straightforward processes, with the final step involving high temperatures. HC is produced by heating a phenolic resin precursor. At the same time, HC@Si is made similarly, but with the addition of 100 nm diameter nano-silicon particles. The phenolic resin precursor is initially subjected to a 1000 °C treatment within a tube furnace under a flowing Ar atmosphere for 3 h. This is followed by purification to eliminate impurities by 1 M HCl (10%, RDH) etching multiple times. Subsequently, 0.9 g of pre-synthesized HC and 0.1 g of commercial nano-silicon (~10 nm, Amita Technologies Inc., Taoyuan, Taiwan) are combined in a beaker. A total of 100 mL of acetone (99%, SIA) is introduced into the crucible and the mixture is subjected to ultrasonic agitation for 4 h. Following sonication, the sample is dried before a final high-temperature treatment at 1000 °C in argon for 3 h, forming HC@Si. The soft carbon (pitch)-coated nano-silicon and phenolic resin (hard carbon) shell composite, HC@Si-P, is synthesized following a procedure similar to that previously described. The key difference occurs in the second synthesis step when 0.4 g of pitch is added to the HC and nano-silicon. The quantity of acetone added to make the mixture is unchanged (100 mL). Following 4 h of ultrasonic agitation, the material is heated to 1000 °C in Ar for 3 h, leading to the formation of HC@Si-P.

### 2.2. Characterizations

Scanning electron microscopy (FE-SEM, JSM-7600F, JEOL Ltd., Akishima, Japan) and high-resolution transmission electron microscopy (HR-TEM, JEM2100, JEOL Ltd.) were utilized to investigate the morphology, particle size, and particle size distribution of various components and composites, including silicon (Si), hard carbon (HC), HC@Si, and HC@Si-P. The corresponding phase purity, crystal structure and structural defects of these samples were investigated using powder X-ray diffraction (XRD, Bruker D2, Billerica, MA, USA), which ranges from 10° to 90° and Raman spectroscopy (ProTrusTech RAMaker, Tainan, Taiwan), which ranges from 300 to 1800 cm^−1^. X-ray photoelectron spectroscopy (XPS), using the JEOL Photoelectron Spectrometer model JPS-9200 (JEOL Ltd.), equipped with monochromatic Al-Kα radiation, was utilized to analyze the surface chemical composition of the samples. This technique involves irradiating the sample with X-rays and measuring the emitted photoelectrons to determine the material’s surface’s elemental composition and chemical states.

### 2.3. Electrode Preparation and Half-Cell Fabrications

The working electrodes were fabricated by combining 80% active materials with 10% carbon black (Super P, Timcal^®^, Shanghai, China) and 10% water-based binder. The binder consisted of 6% carboxymethylcellulose (CMC) and 4% styrene-butadiene rubber (SBR). This blend ensures a stable and efficient electrode by incorporating carbon black for electrical conductivity and a binder to hold the active materials together and improve adhesion to the electrode substrate. The resulting mixture was dissolved in deionized water, forming a slurry uniformly coated onto a 10 μm thick copper foil. The slurry-coated copper foil was then dried at 120 °C for 12 h under vacuum, followed by natural cooling to room temperature to obtain electrodes. The as-prepared electrode was punched into disk-shaped electrodes with a diameter of 14 mm and a thickness of approximately 40 μm. The mass loading of the electrodes was 1.2–1.7 mg cm^−2^. These electrodes were utilized in the assembly of cells for electrochemical measurements. CR2032 coin-type cells were constructed with electrodes formed into circular discs measuring 12 mm in diameter. The assembly process took place inside an Ar-filled glove box, which maintains extremely low levels of water and oxygen (<0.5 ppm). This controlled environment is crucial for preventing contamination and degradation of the cell components, ensuring optimal performance and longevity of the batteries. Lithium metal foil purchased from UBIQ Technology Co., Ltd. (Taoyuan, Taiwan) was used as the counter electrode, separated from the anode by a polypropylene separator. The cell components were crimped together at a pressure of 2.5 MPa. The liquid electrolyte composition consisted of 1 M LiPF_6_ (UBIQ Technology Co., Ltd.) as the lithium salt in a 1:1:1 (by volume) mixture of ethylene carbonate (EC), diethyl carbonate (DEC), and dimethyl carbonate (DMC) solvents, with 5 vol.% fluoroethylene carbonate (FEC) added as an electrolyte additive.

### 2.4. Electrochemical Measurements

Galvanostatic discharge/charge measurements were performed using an AcuTech (McLean, VA, USA) battery testing system (model 750B). This system allowed for programmable constant current and constant voltage operation. The electrochemical measurements of a series of Si/C-based materials were performed within a potential range of 0.01–3.0 V (V vs. Li/Li^+^) at 22 ± 2 °C temperatures. Cyclic voltammograms (CVs) were obtained using a CH Instruments Analyzer (CHI 6273E, Austin, TX, USA) electrochemical workstation. The scan rate used for the measurements was 0.1 mV s^−1^, and the potential range investigated was from 0.01 to 3.0 V. AC impedance measurements were carried out using an AC voltage of 5.0 mV amplitude. The frequency range for the measurements was from 0.01 Hz to 100 kHz. The corresponding electrochemical analyses, such as cycle life, reversible capacity, and rate capability, were evaluated by pouch cells using a multi-channel automated battery cycler. The cells underwent constant discharge currents ranging from 0.1 C to 5.0 C. These cells were initially charged to 4.3 V using a constant current and then discharged down to 2.7 V. For the cycling stability tests, the cells were operated within a voltage range of 2.7 to 4.3 V with a charge and discharge rate of 0.2 C. This approach helps evaluate how well the cells maintain their performance over multiple charge-discharge cycles within the specified voltage range and rate. The pouch cell was constructed using NCM811 cathodes, HC@Si-P anodes, and a polypropylene separator from Foresight Energy Tech., Tainan city, Taiwan. It consisted of stacked cathode and anode electrode sheets measuring 33 mm × 34 mm, separated by the separator. The electrolyte consisted of 1 M LiPF_6_ (99.99%, Aldrich, St. Louis, MO, USA) dissolved in a 1:1:1 (by volume) mixture of ethylene carbonate (EC), diethyl carbonate (DEC), and dimethyl carbonate (DMC) solvents, with 5 vol.% fluoroethylene carbonate (FEC) added as an electrolyte additive. The assembly process took place in a dry room with a dew point temperature of −45 °C, and it took 8 h for the cells to soak in the electrolyte before electrochemical tests were conducted.

## 3. Results and Discussion

Figure 1a presents a schematic illustration summarizing the proposed process for fabricating HC@Si-P nanocomposites. After mixing silicon nanoparticles, hard carbon, and pitch in a beaker, heating the mixture at high temperatures results in hard carbon spheres covered with smaller silicon particles. Hard carbon and silicon surfaces are coated with pyrolyzed carbon from the pitch used in the original mixture. This premise was evaluated using a variety of experimental characterization techniques. Figure 1b shows the X-ray diffraction (XRD) patterns of the as-prepared coated HC@Si-P nanocomposite, alongside the XRD patterns of the individual hard carbon (HC), the uncoated HC@Si composite, and the pristine silicon nanoparticles used in the synthesis. All silicon-containing samples exhibit five distinct diffraction peaks at 28.4°, 47.3°, 56.1°, 69.1°, and 76.4°, corresponding to the (111), (220), (311), (400), and (331) crystal planes of silicon, respectively [51].

In contrast, the XRD pattern of the HC sample shows no sharp Bragg peaks. Instead, it exhibits broad reflections around 23° and 43° 2θ, indicating a highly disordered structure. These broad peaks are also present in the composite samples, suggesting that the carbon structure remains disordered after the composite formation. Furthermore, the absence of additional peaks in the HC@Si-P sample indicates that the carbon coating layer introduced in this material is amorphous. Figure 1c displays the Raman spectra for the carbon-containing samples: HC, HC@Si, and HC@Si-P. These spectra offer detailed information about the structural properties of these materials, highlighting differences in their graphitic and disorder characteristics. The Raman spectra reveal key structural features, such as the intensity ratio of the D and G bands, which can indicate variations in the degree of graphitization and the presence of defects or functional groups in the samples. The G band is characteristic of graphitic carbon materials and signifies the presence of crystalline, ordered sp^2^-bonded carbon. The D-band arises due to structural defects, disorder, or edges within the carbon lattice. It is caused by breathing modes of sp^2^ atoms in rings and activated by the disorder. Our Raman spectra displayed two significant carbon peaks around 1340 and 1590 cm^−1^, corresponding to the disorder-induced D band and the graphitic G band of carbon, respectively. The I_D_/I_G_ ratio for HC@Si-P was approximately 0.84, indicating the level of disorder relative to the graphitic structure. It indicates that incorporating Si slightly increases the ID/IG ratio, suggesting a minor increase in structural disorder. This can be attributed to the interaction between Si and the carbon matrix, which may induce defects and disrupt graphitic domains. However, this controlled level of disorder is beneficial, as it can enhance lithium-ion storage by increasing active sites while maintaining sufficient graphitic regions for electronic conductivity. Scanning electron microscopy (SEM) images provide a visual progression of the synthesis process: Figure 1 shows the pyrolysis of phenolic resin into hard carbon (HC), Figure 2a demonstrates the incorporation of nano silicon particles, and Figure 2b illustrates the carbon coating applied through pitch sintering. The energy-dispersive X-ray spectroscopy (EDS) mapping shown in Figure 2c–e illustrates the unique spatial arrangement of carbon and silicon elements within the HC@Si-P composite. This technique allows us to visualize how carbon and silicon are distributed throughout the composite material, highlighting their specific locations and how they interact within the structure. Carbon is predominantly found in the large, micron-sized HC spheres. At the same time, silicon is concentrated in the smaller particles attached to these spheres. The EDS spectrum and analysis in Figure 2f provide a carbon-to-silicon ratio that aligns with the proportions of the starting materials used in the composite preparation, confirming the successful integration of the components.

Transmission electron microscopy (TEM), as illustrated in Figure 3, was utilized to examine the structural characteristics in greater depth. Figure 3a clearly illustrates that the porous nanoscale carbon coating uniformly encapsulates the HC@Si, forming a core-shell nanocomposite designated HC@Si-P [52,53]. In Figure 3b, a high-resolution transmission electron microscope (TEM) image of the HC@Si-P nanocomposites shows a carbon coating layer that is somewhat irregular but robust, with a thickness of about 5 nm. This amorphous carbon layer uniformly envelops the surface of the HC@Si composite. The image highlights the consistency of the coating, which is crucial for the material’s performance and stability, suggesting effective integration of the carbon coating with the core composite. This flexible nanoscale carbon coating is essential as it accommodates the volume expansion of nanoscale silicon while facilitating easy access of the electrolyte to the silicon.

Additionally, the inset in Figure 3b shows a selected area electron diffraction (SAED) pattern, confirming the polycrystalline nature of the HC@Si-P nanocomposite [54]. The diffraction rings observed in the SAED pattern correspond well with the cubic structure of silicon, aligning with the powder X-ray diffraction (XRD) patterns presented in Figure 1. Furthermore, Figure 2b highlights the core-shell structure of the nanocomposites, with the TEM image showing an interplanar distance of 0.31 nm, consistent with the (111) plane of silicon. Energy-dispersive X-ray spectroscopy (EDS) mapping results, shown in Figure 3c–f, further corroborate the uniform distribution of silicon and carbon within the nanocomposites, emphasizing the homogeneous nature of the material. X-ray photoelectron spectroscopy (XPS) was employed to analyze the chemical states of the elements within the HC@Si-P nanocomposites. Figure 4a displays a broad survey scan across various binding energies, confirming the presence of oxygen alongside the expected silicon and carbon components in the HC@Si-P nanocomposite. High-resolution XPS spectra for HC@Si-P are shown in Figure 4b–d. In Figure 4b, the deconvoluted peaks at 103.5 eV, 101.8 eV, and 99 eV in the Si 2p region are associated with Si–O, Si–C, and Si–Si bonds, respectively, indicating the presence of different silicon species within the HC@Si-P structure. Figure 4c reveals XPS peaks in the O 1s region, with 533.5 eV and 532.4 eV corresponding to C–O and C=O bonds, suggesting the incorporation of oxygen-related functionalities. Additionally, Figure 4d shows peaks at 284.6, 286.4, and 288.7 eV in the C1s region, which correspond to sp2–C, C=O, and O–C=O bonds, respectively, confirming the presence of various carbon-based functional groups within the nanocomposite [55]. These results provide a detailed understanding of the chemical composition and bonding environment in the HC@Si-P nanocomposites [56].

Cyclic voltammetry (CV) curves obtained for the first three cycles for pristine HC, HC@Si, and HC@Si-P are shown in Figure 5a,d,g, respectively. During the initial reduction phase, a small peak was observed at around 0.5–0.7 V, likely due to the breakdown of the electrolyte and the creation of a solid electrolyte interphase (SEI) film on the electrode surface. Additionally, a prominent peak close to 0.1 V and a broader, more gradual peak spanning from 0.2 to 1.5 V corresponds to the plateau and the sloped regions seen in the charge/discharge curves, respectively, which indicates that the initial reduction process involves complex electrochemical interactions where the SEI formation and the different voltage regions reflect distinct phases of the charge/discharge behavior [57]. The potential difference between the oxidation and reduction peaks (ΔE = Epc − Epa) showed a more significant disparity when the material underwent significant polarization. This polarization phenomenon happened because the transport of electrons or ions was slower, which required a more significant potential difference to drive the process. From a comparison of Figure 5b,e,h, HC@Si-P displayed superior overlap at varying scanning rates, indicating excellent cyclic stability. Furthermore, the diffusion coefficients of HC, HC@Si, and HC@Si-P, as depicted in Figure 5c,f,i, followed the Randles–Sevcik Equation (1):(1)$$I_{p}=2.69\times 10^{5}n^{3/2}AD^{1/2}Cv^{1/2}$$
*I_p_* represents the peak current, *n* is the charge transfer number, *A* denotes the active electrode area, *D* is the lithium-ion diffusion coefficient, *C* is the molar concentration of Li^+^ in the anode, and *ν* is the scan rate [58]. The diffusion coefficients, *D*_Li+_, were calculated to be 6.4 × 10^−10^ cm^2^ s^−1^, 3.3 × 10^−10^ cm^2^ s^−1^, and 9.7 × 10^−10^ cm^2^ s^−1^ for HC, HC@Si, and HC@Si-P, respectively, which indicated that the diffusion coefficient of HC@Si-P was higher than those of the other samples, corroborating the results of the CV tests.

To further evaluate the rate performance of the Si/HC electrode, the first three cycles were displayed to accurately assess the coulombic efficiency (CE) of the first cycle, as shown in Appendix A. Following this, we conducted rate capability tests on the three samples, with the results in Figure 6a. The discharge-specific capacity of the HC@Si-P electrode exhibited a decreasing trend as the current density increased, starting from 560 mAh/g at 0.1 A/g and dropping to 215 mAh/g at 5 A/g. Remarkably, when the current density was reduced to 0.1 A/g, the electrode’s capacity recovered to 541 mAh/g. This demonstrates the electrode’s excellent recovery capacity after rapid charging and discharging cycles. This recovery can be attributed to two main factors: the formation of a stable solid electrolyte interphase (SEI) layer on the surface of the core-shell nanocomposite and the improved electrical conductivity provided by the carbon frameworks. The SEI layer is a protective barrier that stabilizes the electrode material during cycling.

At the same time, the carbon framework enhances overall electrical conductivity, promoting efficient charge transfer. To further elucidate the differences in rate capability among the three samples, we employed four-point probe measurements to determine the resistance and conductivity of the Si, HC, HC@Si, and HC@Si-P electrodes, with the results illustrated in Appendix A. The HC@Si-P composite electrode exhibited a significantly lower resistance of 0.01283 ohms than the other electrodes. Correspondingly, the room temperature electronic conductivity of the HC@Si-P composite electrode was notably higher (7.8 × 10^4^ S cm^−1^) than the conductivity of the other electrodes [59]. In addition to conducting cyclic voltammetry (CV) and four-point probe tests, we also performed electrochemical impedance spectroscopy (EIS) measurements on the electrodes. To underscore the benefits of the pitch coating process, we delved into a detailed examination of the electrochemical behavior and resistance characteristics. By employing electrochemical impedance spectroscopy (EIS) measurements, we could thoroughly analyze the impedance and various electrical properties. This comprehensive analysis highlighted the improved performance of the electrodes. It demonstrated how pitch coating significantly enhances their efficiency and durability. This approach provides a clearer understanding of how the coating contributes to superior electrode functionality. Figure 6b illustrates the average capacity retention of HC, HC@Si, and HC@Si-P electrodes across multiple rate cycles as the current density varies. The HC electrode exhibited a steady decline in capacity retention as the current density increased, ultimately retaining only 22.1% of its initial capacity at a high current density of 5 A g^−1^. In contrast, the HC@Si-P electrode exhibited significantly improved performance, retaining 38.4% of its initial capacity at 5 A g^−1^. Furthermore, when the current density was reduced to the initial value of 0.1 A g^−1^, the HC@Si-P electrode maintained an impressive average retention rate of 96.6%.

The performance enhancement of the HC@Si-P electrode is notably superior to cells using either HC or HC@Si electrodes. This improvement is primarily attributed to the carbon coating derived from pitch, which significantly enhances the electrical conductivity of the HC@Si composite. Carbon coating not only improves electrical connectivity but also boosts the overall efficiency and stability of the electrode across various applications. Appendix A provides a detailed comparison of the charge–discharge cycles (potential vs. capacity) as a function of current density and cycle number for each electrode, further emphasizing the superior performance of the HC@Si-P electrode. Galvanostatic charge/discharge cycling tests were performed on the electrodes within a voltage range of 0.01 to 3.0 V, starting at a current density of 0.1 A/g for the first three cycles, followed by subsequent tests at 1.0 A g^−1^. These tests aimed to evaluate the cycling performance of a hard carbon (HC) electrode and various HC core-shell nanocomposite electrodes, with and without additional pitch treatment. Figure 6c shows that the HC@Si electrode initially exhibited a charge capacity of 488 mAh/g. However, it experienced rapid capacity degradation, eventually dropping to below 250 mAh/g after 200 cycles. This capacity decline is likely attributed to the substantial volume changes in silicon during the lithiation and delithiation processes, leading to structural instability in the electrode.

In contrast, the HC@Si-P material, which contains 10 wt% silicon, demonstrated significantly better cycling stability. It retained a reversible specific capacity of approximately 400 mAh/g over 200 cycles. This enhanced performance can be attributed to the carbon coating derived from pitch, which effectively manages the expansion of silicon during cycling and reduces internal stress within the electrode. This coating effectively addresses the issues of silicon’s volume change, leading to better stability over many charge and discharge cycles. The minimal capacity fluctuation observed in Figure 6c (green) could be attributed to the structural and interfacial changes occurring during prolonged cycling. Additionally, minor variations in lithium-ion accessibility to active sites and gradual stabilization of the electrode structure may contribute to these fluctuations. Appendix A illustrates the galvanostatic discharge-charge profiles for pristine HC, HC@Si, and HC@Si-P at a current density of 1 A/g across the 1st, 5th, 10th, 50th, 100th, and 200th cycle. During the discharge and charge processes, plateaus at 0.1 V and 0.4 V, respectively, correspond to the lithiation and delithiation of the silicon. Notably, the HC@Si-P electrode achieved an initial charge capacity of 749 mAh/g with a first coulombic efficiency (CE) of 78.9% (as shown in Figure 6c). The initial discharge curve, indicative of the lithiation of crystalline silicon, features a lengthy and flat discharge plateau at 0.08 V. By the fifth cycle, the coulombic efficiency of HC@Si-P had reached an impressive 99.2%, surpassing that of both HC and HC@Si. Appendix A presents a detailed analysis of the capacity contributions during different stages of the lithiation and delithiation processes across multiple cycles for the Si, HC, HC@Si, and HC@Si-P electrodes. After 200 cycles, the HC@Si-P electrode exhibited slope/plateau lithiation capacities at the 1st, 5th, 10th, 50th, and 100th cycles, measured as 263/486, 273/206, 273/189, 250/163, and 236/166 mAh/g, respectively. By the 200th cycle, it maintained 220/174 mAh/g capacities during the charge/discharge processes. The contribution of the slope capacity for HC@Si-P was 35.2% in the initial cycle, increasing to 57.0%, 59.1%, and 55.8% in the 5th, 10th, and 200th cycles, respectively, when compared to the first cycle. For the HC electrode, the plateau capacity contribution was 40.1%, 34.7%, 32.5%, and 38.1% at the 1st, 5th, 10th, and 200th cycles, respectively. In comparison, HC@Si displayed contributions of 38.8%, 30.2%, 29.5%, and 38.8% at the same cycles. The consistent slope and plateau contributions observed in the HC@Si-P electrode over 200 cycles indicate that lithium ions predominantly move through the charge gradient from the surface defect sites during the lithium diffusion process. This indicates that the electrode’s structure, particularly the pitch-derived carbon coating, is critical in maintaining stability and performance during cycling [60].

The steep slope in discharge capacity indicates that most lithium-ion storage occurs at the surface defect sites of the material. These defects provide additional sites for lithium-ion interactions. The small plateau observed at lower voltages (below 0.01 V) may be due to residual oxygen-containing groups and an incomplete graphitic structure. The oxygen functional groups not fully incorporated into the graphitic lattice can form strong bonds with lithium ions. This interaction enhances the material’s ability to store lithium ions at these sites, influencing overall discharge characteristics. It suggests that the primary mechanism for lithium-ion storage is adsorption onto these chemical functional groups rather than insertion into the disordered carbon layers. In the case of the HC@Si-P material, the lithiation (charging) capacity exceeds its delithiation (discharging) capacity. This implies that under overpotential conditions, the release of lithium ions is limited, making it challenging for them to be inserted into the graphitic layer structure. This observation suggests that the lithium-ion storage mechanism in HC involves a combined process of “adsorption–intercalation”, where lithium ions first adsorb onto the surface before possibly intercalating into the material’s structure. A typical Nyquist plot, obtained from electrochemical impedance spectroscopy (EIS), exhibits semi-circular curves at high and intermediate frequencies and a slanted line at low frequencies. The position along the frequency and the intercept represents the ohmic resistance (R_S_) arising from the combined resistance of the electrode, electrolyte, and separator within the battery. A semicircle at medium frequencies corresponds to the impedance of the solid electrolyte interface (R_SEI_).

In contrast, the semicircle at lower frequencies is associated with charge transfer resistance (R_ct_) or polarization resistance. A downward trend is often observed at low frequencies, reflecting ion diffusion within the battery. In the low-frequency range, a plot of real impedance versus the reciprocal of the square root of the frequency can be generated. The slope of this plot enables the calculation of the diffusion coefficient (*D*) using the relevant Equation (2) [61]:(2)$$D=\frac{R^{2}T^{2}}{2A^{2}n^{4}F^{4}C^{2}\sigma_{W}^{2}}$$

In this equation, *R* represents the gas constant, *T* is the temperature, *n* is the number of electrons per molecule involved in the oxidation process, *A* denotes the surface area of the electrode, *F* is the Faraday constant, *C* is the lithium-ion concentration, *σ_w_* is the Warburg factor, and *D* represents the diffusion coefficient. According to the formula, a smaller impedance-frequency reciprocal square root relationship slope corresponds to a more significant diffusion coefficient.

The Nyquist plots generated from the EIS measurements for each half-cell containing the respective electrodes are shown in Figure 7a. The impedance values obtained by fitting the Nyquist plots to equivalent circuits are presented in Appendix A. A strong correlation exists between the specific surface area (SSA) and the R_SEI_; thus, if SSA is high, R_SEI_ also tends to increase. HC@Si-P exhibits lower impedance compared to the other samples after 2.5 cycles. The lower SSA encourages the creation of a SEI film. At the same time, huge pores may result in excessive electrolyte uptake and undesirable side reactions during the electrochemical process. Plots of Z′ vs. ω^−0.5^ for each of the electrodes, HC, HC@Si, and HC@Si-P, as calculated from the linear, low-frequency region of the Nyquist plots, are shown in Figure 7b.

The corresponding diffusion coefficients for HC, HC@Si, and HC@Si-P are 7.7 × 10^−13^, 1.4 × 10^−12^, and 1.8 × 10^−12^ cm^2^ s^−1^, respectively. These results suggest that HC@Si-P demonstrated the highest diffusion coefficient. The pore characteristics of HC can influence lithium storage by decreasing the diffusion distance for lithium ions and increasing the number of active sites. Figure 8a–d presents SEM images of pristine Si and HC@Si-P electrodes before and after 200 cycles. Initially, both types of electrodes displayed a dense surface morphology. After 200 cycles, the pristine Si electrode, lacking HC incorporation and pitch coating, exhibited numerous cracks due to the significant volume expansion of Si during the charge and discharge processes, as shown in Figure 8b. In contrast, the HC@Si-P electrode, shown in Figure 8d, maintained a uniform and compact surface structure without cracks, indicating that the integration of HC and pitch coating significantly enhances the structural stability of the electrode during cycling. Furthermore, as depicted in Figure 9, the pristine Si electrode undergoes a substantial volumetric expansion of 433.0% after cycling. On the other hand, the HC@Si-P electrode experiences a much lower volumetric expansion of 145.5%. It suggests that the pitch coating effectively mitigates the volume expansion of Si, enhancing the electrode’s mechanical stability and overall performance during charge and discharge cycles. The performance difference between the two electrodes can be attributed to the pitch coating’s capacity to form a robust and flexible matrix, accommodating the expansion and contraction of Si. By buffering the volume changes, the pitch-coated structure reduces the risk of cracking and structural degradation, which are common issues in Si-based anodes. This improvement is essential for enhancing the longevity and efficiency of lithium-ion batteries [62,63]. Subsequently, we assembled and tested pouch cells using the anode material derived from HC@Si-P and a cathode material of NCM-811 (LiNi_0.8_Co_0.1_Mn_0.1_O_2_) (Figure 10). Remarkably, the coulombic efficiency surpassed 99.0%. The capacity retention remained at 56.7% even after 200 cycles. When comparing the electrochemical performance of the composition-optimized HC@Si-P to other studies listed in Appendix A, it is evident that the synthesized HC@Si-P composite holds significant promise for use in Li-ion battery applications.

The schematic representation is shown in Figure 11. illustrates the comparative structural changes likely occurring in HC@Si-P and bare Si nanoparticles during cycling. In the case of HC@Si-P, while the silicon component undergoes some volume changes, the carbon coating derived from pitch effectively reduces these significant fluctuations, thereby improving structural stability. This improvement results in more stable cycling performance. In comparison, bare silicon nanoparticles undergo repeated volume expansion and contraction during charge and discharge cycles, which causes them to fragment and form an unstable solid electrolyte interphase (SEI) film. This instability in the SEI film leads to degraded cycling performance. Essentially, the enhanced stability helps to mitigate the mechanical stress and degradation associated with volume changes, thereby maintaining better performance over time. Given that a large portion of silicon nanoparticles in HC@Si composites remains uncoated with carbon, it is understandable that HC@Si and other previously reported Si/C composite electrodes exhibit suboptimal cycling stability.

## 4. Conclusions

In this study, we utilize hard carbon as the core and apply a nano-silicon/pitch-derived carbon coating as the shell. We have successfully synthesized HC@Si-P composite anode materials that improve in cycling compared to similar electrodes without this external coating. Our composition-optimized has exhibited exceptional cycling stability, enduring up to 200 cycles at 1 A/g with only a minimal 0.16% capacity decay per cycle. These promising results underscore the potential of our Si/C composite anode material as a viable candidate for high-rate, high-performance lithium-ion batteries, addressing many of the longevity issues associated with operating silicon anodes at high current densities and bringing us closer to realizing advanced energy storage solutions.

## Figures and Tables

**Figure 1 nanomaterials-15-00455-f001:**
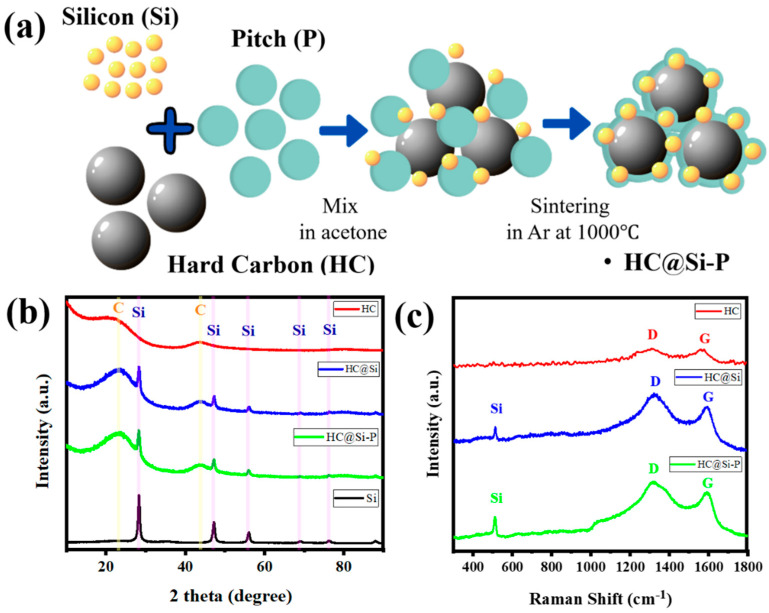
(**a**) A schematic showing the proposed evolution of structures during the synthesis of HC@Si−P composites; (**b**) XRD patterns of HC, HC@Si, HC@Si−P, and Si; (**c**) Raman spectra of HC, HC@Si, and HC@Si−P.

**Figure 2 nanomaterials-15-00455-f002:**
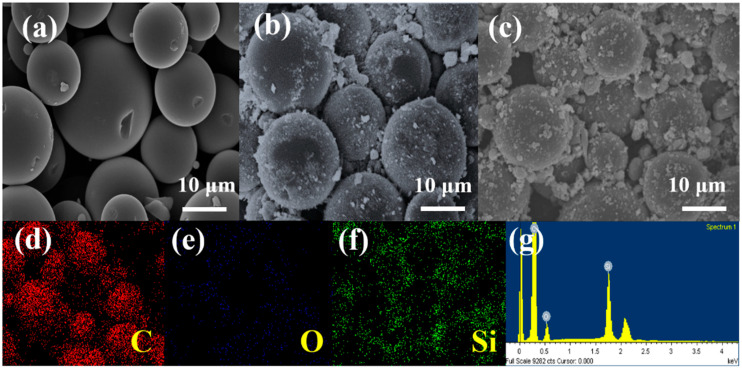
SEM images of (**a**) HC, (**b**) HC@Si, and (**c**) HC@Si−P; (**d**), carbon, (**e**) oxygen, and (**f**) silicon EDS elemental maps taken from the sample of HC@Si−P; (**g**) An EDS spectrum and the corresponding quantitative analysis of HC@Si−P.

**Figure 3 nanomaterials-15-00455-f003:**
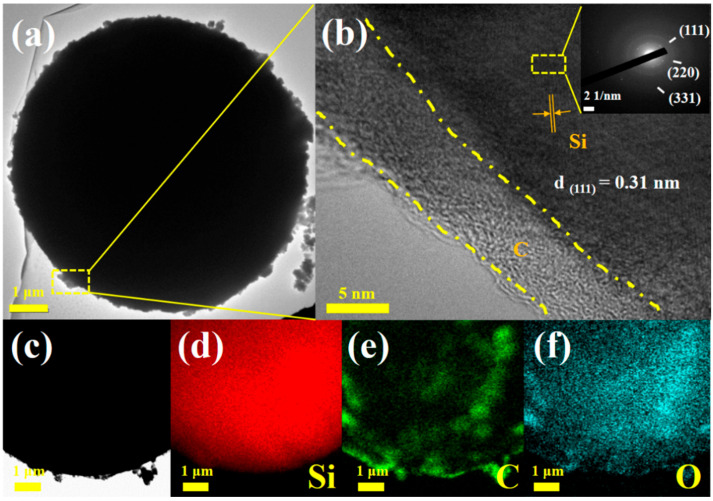
(**a**–**c**) HRTEM images of HC@Si-P; (**d**–**f**) EDS elemental mapping images of HC@Si-P.

**Figure 4 nanomaterials-15-00455-f004:**
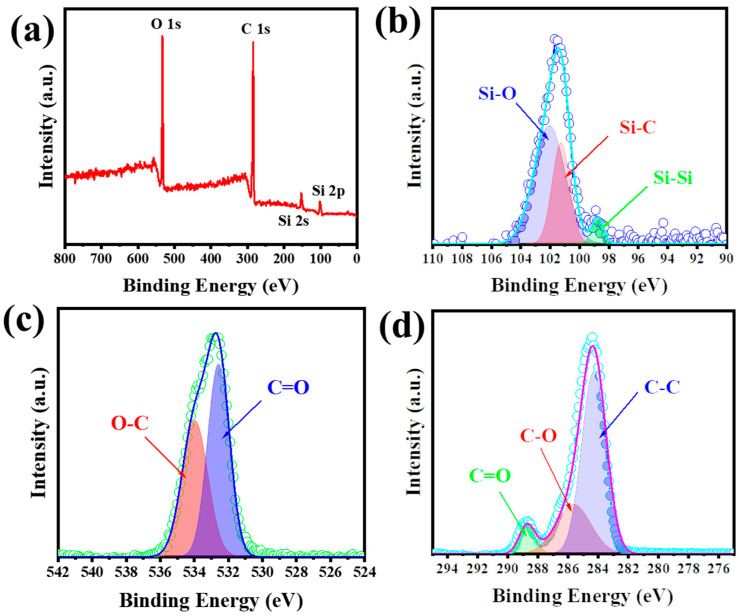
The XPS survey spectra of (**a**) HC@Si−P, (**b**) Si 2p, (**c**) O 1s, and (**d**) C 1s.

**Figure 5 nanomaterials-15-00455-f005:**
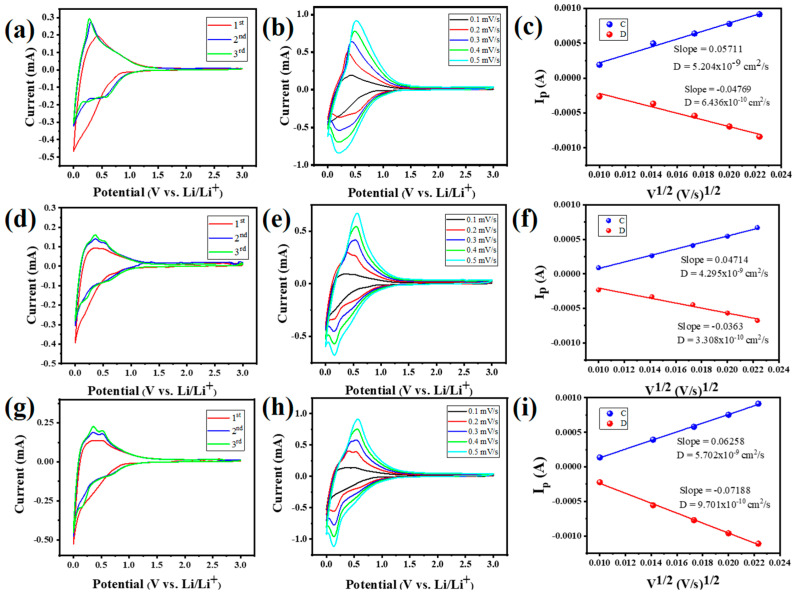
Cyclic voltammetry (CV) curves for (**a**) HC, (**d**) HC@Si, and (**g**) HC@Si−P electrodes were taken at a scan rate of 0.1 mV s^−1^, for the initial three cycles at various scan rates (**b**,**e**,**h**) and diffusion coefficient (**c**,**f**,**i**).

**Figure 6 nanomaterials-15-00455-f006:**
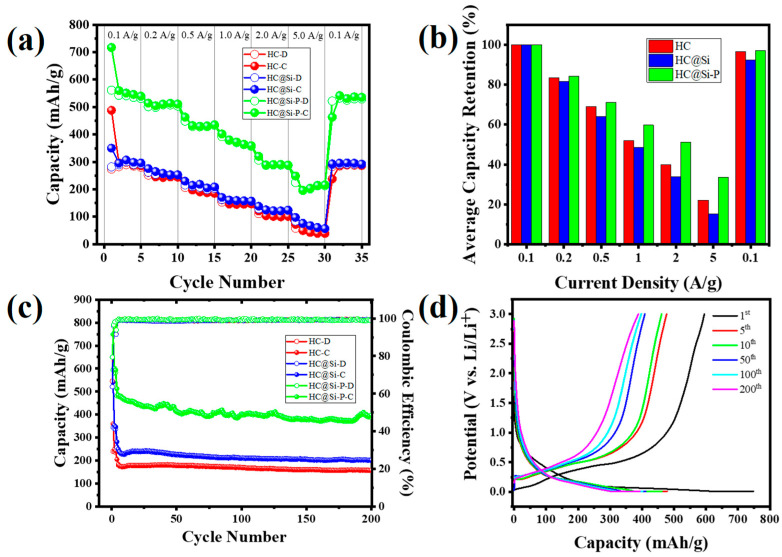
(**a**) Rate capability of HC, HC@Si, and HC@Si-P electrodes under varying current density from 0.1 to 5.0 A g^−1^; (**b**) A histogram comparing the average reversible capacity of the HC, HC@Si, and HC@Si-P electrodes across variable applied current densities. (**c**) Cycling performance of HC, HC@Si, and HC@Si-P at 1.0 A/g over 200 cycles. The first three cycles were performed at a lower current density of 0.1 A g^−1^. (**d**) Charge/discharge profiles of HC@Si-P measured at a current density of 1.0 A g^−1^, with the exception of the first cycle, which was performed at 0.1 A g^−1^.

**Figure 7 nanomaterials-15-00455-f007:**
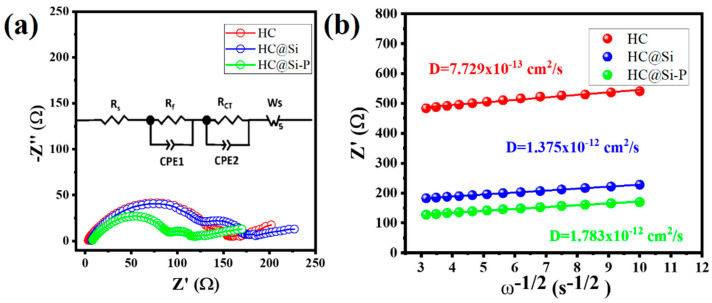
(**a**) Nyquist plots taken from EIS measurements made on the cells containing each of the respective electrodes (red: HC; blue: HC@Si; green: HC@Si-P) after 2.5 cycles; (**b**) A plot of Z′ vs. ω^−0.5^ for each of the electrodes, HC, HC@Si, and HC@Si-P, as calculated from the linear, low-frequency region of the Nyquist plots. The diffusion coefficients, D, were calculated from the gradients of the linear plots.

**Figure 8 nanomaterials-15-00455-f008:**
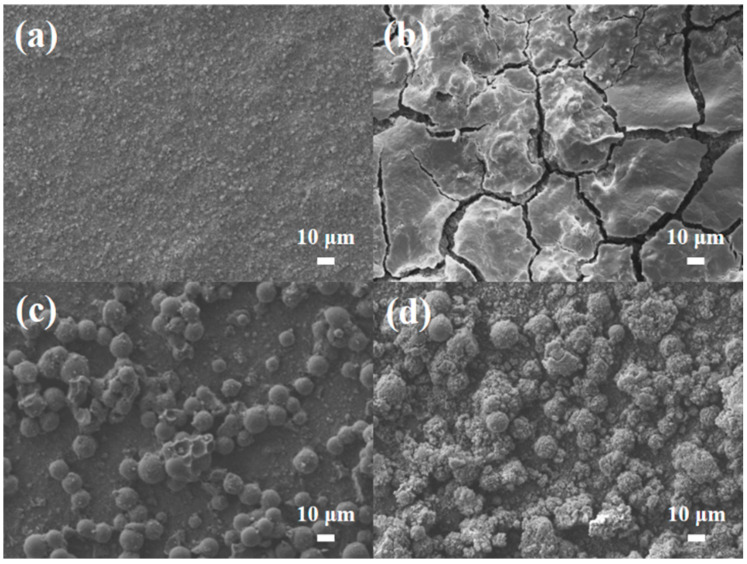
Top view SEM images for (**a**) Si electrode, (**c**) HC@Si-P electrode, (**b**,**d**) Si and HC@Si-P electrode after 200 cycles.

**Figure 9 nanomaterials-15-00455-f009:**
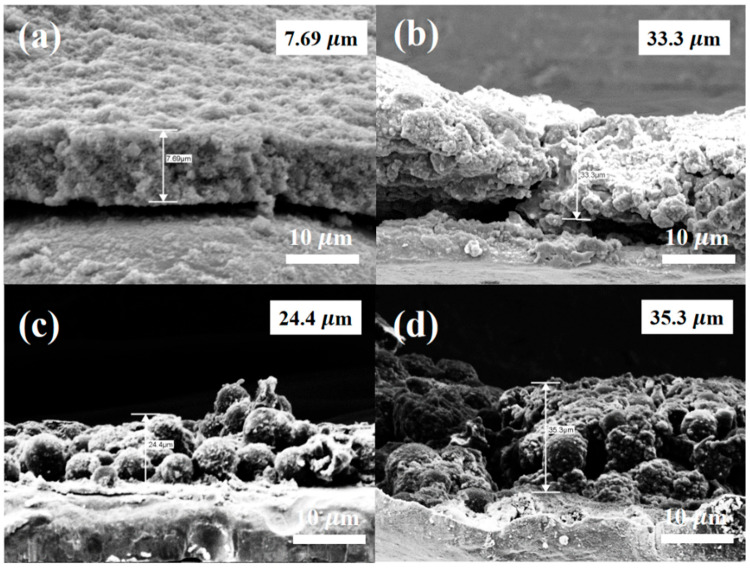
Cross-section SEM images for (**a**) Si electrode, (**c**) HC@Si-P electrode, (**b**,**d**) Si and HC@Si-P electrode after 200 cycles.

**Figure 10 nanomaterials-15-00455-f010:**
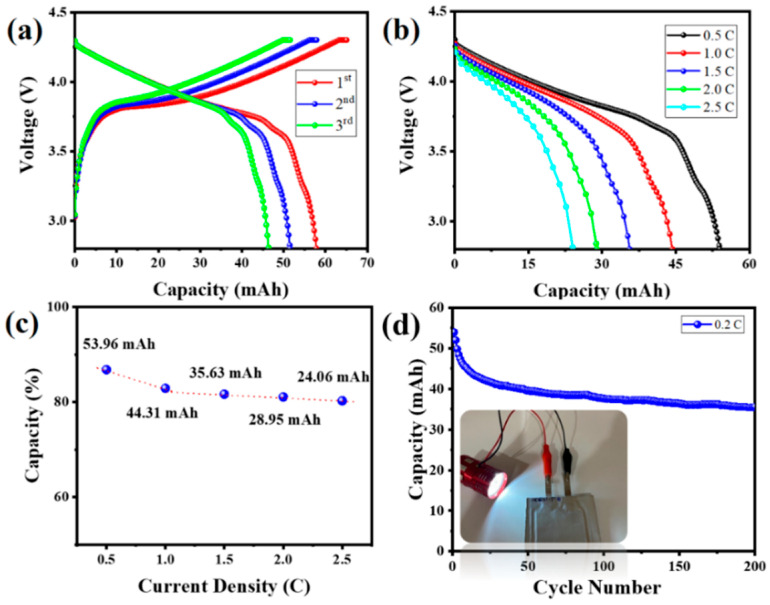
(**a**) The charge/discharge curves of an NCM811/HC@Si-P pouch cell for the first three cycles; (**b**) the charge/discharge curves of the pouch cell cycled at current densities of 0.5, 1.0, 1.5, 2.0, and 2.5 C, (**c**) rate performance of the pouch cell in terms of % capacity vs current density; (**d**) extended cycling performance of the pouch cell at 0.2 C for 200 cycles.

**Figure 11 nanomaterials-15-00455-f011:**
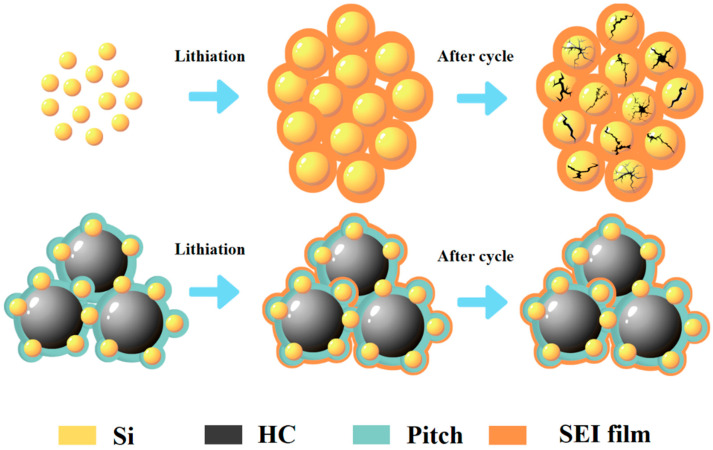
Schematic illustrations comparing the influences of the cycling process on a pristine silicon electrode vs. a composite HC@Si-P electrode.

## Data Availability

The original contributions presented in this study are included in the article/Appendix A. Further inquiries can be directed to the corresponding author(s).

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
