# Peer review of "Silicon/Hard Carbon Composites Synthesized from Phenolic Resin as Anode Materials for Lithium-Ion Batteries"

_nanomaterials, 2025, doi:10.3390/nano15060455_

Round 1

Reviewer 1 Report

Comments and Suggestions for Authors

The authors have presented the paper entitled " Silicon/hard carbon composites synthesized from phenolic resin as anode materials for lithium-ion batteries"

The manuscript is very well organized is easy to read and is very interesting.

The approach of the paper is really interesting and it can contribute greatly to the field.

I have no many comments, the paper is really nice. Perhaps the only note is : 

Why do the authors didn't try the device under different temperature conditions?

Do the authors have measure BET studies? This could be very interesting

Why do the performance is similar and not enhanced? This performance was expected at least by me while reading the paper? Can you explain further on this matter?

Author Response

Reviewer: 1

Comments:

The authors have presented the paper entitled " Silicon/hard carbon composites synthesized from phenolic resin as anode materials for lithium-ion batteries" The manuscript is very well organized is easy to read and is very interesting. The approach of the paper is really interesting and it can contribute greatly to the field. I have no many comments, the paper is really nice. Perhaps the only note is :

Response:

Thanks to the reviewer for the general comment on the manuscript and appreciation for the performance of the work. Our comprehensive, point-by-point responses to their comments are provided below.

  1. Why do the authors didn't try the device under different temperature conditions?

Response:

We sincerely appreciate the reviewer's insightful comment. In this study, our primary objective was to develop and evaluate the electrochemical performance of the HC@Si-P composite as a potential anode for Li-ion batteries, focusing on composition optimization to enhance cyclic stability. While we acknowledge the importance of investigating temperature-dependent performance, our current work aimed to establish the fundamental electrochemical properties of the material under standard testing conditions. However, we agree that temperature variations can significantly influence performance, and we consider this an important avenue for future research. We genuinely appreciate this valuable suggestion and will consider it for our future studies.

We hoped that the reviewer would be satisfied with our responses for publication.

  1. Do the authors have measure BET studies? This could be very interesting

Response:

Thank you for the reviewer’s comments. In this study, we did not perform BET measurements, as our primary focus was optimizing the composition and evaluating the electrochemical performance of the HC@Si-P composite anode. However, we agree that BET surface area analysis could provide additional insights into the material's porosity and surface characteristics, which may further support our findings. We truly appreciate this thoughtful suggestion and will consider it in future investigations to deepen our understanding of the material properties.

We hoped that the reviewer would be satisfied with our responses for publication.

  1. Why do the performance is similar and not enhanced? This performance was expected at least by me while reading the paper? Can you explain further on this matter?

Response:

We sincerely appreciate the reviewer's thoughtful observation. The comparison table (Table. R1) shows that the HC@Si-P composite anode demonstrates enhanced cycling stability and initial Coulombic efficiency (ICE) compared to several reported silicon-based anodes. Our as-prepared anode material maintains 391 mAh/g capacity after 200 cycles at 1.0 A/g, highlighting its long-term stability. While some literature-reported materials may show higher capacities, they often operate at lower current densities and limited cycle life. Our work prioritizes a balance between stability and capacity, making the HC@Si-P composite a promising candidate for practical applications.

Table R1. Comparison of electrochemical performance with reported Si/HC as anode materials for Lithium-ion batteries.

Samples

*ICE

After nth cycles

Capacity

(mAh/g)

Current rate

(A/g)

Ref.

HC@Si-P

79%

200

391

1.0

This work

Si/C

57%

50

678

0.1

[1]

Si@C

68%

500

868

0.1

[2]

Si@C-pitch

74%

100

629

0.5

[3]

Si@SiOx@C

64%

50

770

0.2

[4]

SGC

79%

100

1526

0.25

[5]

Si@10C

72%

500

1006

0.5

[6]

Si@C@v@CNTs

76%

100

912

0.1

[7]

Si@Void@NC

74%

400

475.1

0.5

[8]

Si/C-AG

64%

200

445

0.5

[9]

Si@hNC

61%

100

735

0.2

[10]

*ICE: Initial Columbic Efficiency

We hoped that the reviewer would be satisfied with our responses for publication.

Reviewer 2 Report

Comments and Suggestions for Authors

1.    The introduction section is long; please consider restructuring it to improve conciseness and clarity.
2.    The study presents an interesting approach to synthesizing silicon/hard carbon composites from phenolic resin for lithium-ion battery anodes. However, its novelty should be clearly stated, especially in comparison with existing silicon/carbon composites.
3.    Some references are outdated; incorporating recent literature on batteries and Si/C composites for LIB anodes would strengthen the study, such as: Nanoscale, 2025, 17, 6049-6057 (https://doi.org/10.1039/D4NR04463K); ACS Appl. Mater. Interfaces 2024, 16, 26 (https://doi.org/10.1021/acsami.4c00835), 33294–33306; …
4.    Ensure correct typo sentences,’ Introduction: Lithium-ion Lithium-ion batteries (LIBs); ensure that the correct figure number and sub-label are referenced properly, such as SEM Figures.
5.    The porosity of the composite should be discussed, as it influences the buffering effect against Si expansion.
6.    What about the ID/IG ratio for HC@Si and HC in Raman analysis? The authors should clearly report and compare the ID/IG values and discuss how the Si incorporation affects the disorder/graphitization of HC and how this influences electrochemical performance.

Comments on the Quality of English Language

The English can be improved to more clearly express the research.

Author Response

Reviewer: 2

  1. The introduction section is long; please consider restructuring it to improve conciseness and clarity.

Response:

We sincerely thank the reviewer for their insightful comments and suggestions, which have significantly improved the clarity and comprehensiveness of our manuscript. We sincerely appreciate your comment regarding the length and structure of the introduction section. We understand that a more focused and streamlined introduction would enhance the readability and impact of our work. The revised introduction section has significantly improved the conciseness and clarity, making it more effective in highlighting the importance and relevance of our work.

  1. The study presents an interesting approach to synthesizing silicon/hard carbon composites from phenolic resin for lithium-ion battery anodes. However, its novelty should be clearly stated, especially in comparison with existing silicon/carbon composites.

Response:

We thank the reviewer for this important point. The novelty of our work lies in the strategic design of a core-shell structure, where hard carbon derived from phenolic resin serves as a stable core, and nano-silicon with a pitch coating forms the shell. This composition-optimized HC@Si-P composite anode enhances cycling stability and initial Coulombic efficiency (ICE) compared to many previously reported silicon/carbon composites. Unlike conventional Si/C composites, our approach leverages the synergistic effect of pitch coating and hard carbon to mitigate silicon’s volume expansion, ensuring prolonged cycling performance at a practical current density. The comparison table further highlights these improvements. We genuinely appreciate the reviewer’s insightful comment and will emphasize this novelty explicitly in our revised manuscript.

We hoped that the reviewer would be satisfied with our responses for publication.

  1. Some references are outdated; incorporating recent literature on batteries and Si/C composites for LIB anodes would strengthen the study, such as: Nanoscale, 2025, 17, 6049-6057 (https://doi.org/10.1039/D4NR04463K); ACS Appl. Mater. Interfaces 2024, 16, 26 (https://doi.org/10.1021/acsami.4c00835), 33294–33306; …

Response:

We thank the reviewer for recommending valuable papers. We have read them, and they are relevant to our study. Therefore, we would like to cite them in an appropriate sentence on pages 1 and 5  of the revised manuscript.

  1. Chen, J; Wang, X; Deng, Z; Kim, E-M; Jeong, S-M. Facile synthesis of Si/C composites for high-performance lithium-ion battery anodes. Nanoscale, 2025,17, 6049-6057.
  2. Kumar, D-R; Kanagaraj, I; Sukanya, R; Karthik, R; Hasan, M; Thalji, M-R; Dhakal-G, Milton, M; Prakash, A-S; Shim, J-J. Ti3C2Tx Filled in EMIMBF4 Semi-Solid Polymer Electrolytes for the Zinc–Metal Battery. ACS Appl. Mater. Interfaces 2024, 16 (26), 33294-33306.

We hoped that the reviewer would be satisfied with our responses for publication.

  1. Ensure correct typo sentences,’ Introduction: Lithium-ion Lithium-ion batteries (LIBs); ensure that the correct figure number and sub-label are referenced properly, such as SEM Figures.

Response:

We appreciate the reviewer's careful observation of the manuscript. We have thoroughly reviewed the manuscript and corrected the typographical errors, including the duplication in "Lithium-ion batteries (LIBs)." Additionally, we have ensured that all figure numbers and sub-labels, such as those in the SEM figures, are referenced correctly in the revised manuscript.

We hoped that the reviewer would be satisfied with our responses for publication.

  1. The porosity of the composite should be discussed, as it influences the buffering effect against Si expansion.

Response:

We thank the reviewer for raising this important point. In our study, the hard carbon (HC) derived from phenolic resin is a mechanically robust yet structurally adaptable core, which inherently accommodates silicon’s volume expansion. Additionally, the pitch coating plays a crucial role in reinforcing structural integrity while preventing excessive porosity that could lead to unstable solid-electrolyte interphase (SEI) formation and electrolyte decomposition. While we acknowledge that porosity can provide additional buffering, excessive porosity may also reduce the electrode’s volumetric energy density and compromise electrical conductivity. Our optimized HC@Si-P composite anode balances structural stability and electrochemical performance, achieving superior cycling stability without excessive porosity. We have included a discussion on this aspect in the revised manuscript.

We hoped that the reviewer would be satisfied with our responses for publication.

  1. What about the ID/IG ratio for HC@Si and HC in Raman analysis? The authors should clearly report and compare the ID/IG values and discuss how the Si incorporation affects the disorder/graphitization of HC and how this influences electrochemical performance.

Response:

Thanks for your thoughtful and constructive comment regarding Raman's analysis. The ID/IG ratio is a crucial parameter for evaluating the disorder and graphitization degree of hard carbon (HC) shown in Fig. R1. We have now carefully analyzed the ID/IG values for HC and HC@Si from the Raman spectra and included them in the revised manuscript. Our results indicate that incorporating Si slightly increases the ID/IG ratio, suggesting a minor increase in structural disorder. This can be attributed to the interaction between Si and the carbon matrix, which may induce defects and disrupt graphitic domains. However, this controlled level of disorder is beneficial, as it can enhance lithium-ion storage by increasing active sites while maintaining sufficient graphitic regions for electronic conductivity. The balance between disorder and graphitization plays a critical role in achieving stable cycling performance, which is evident in the electrochemical results of the HC@Si-P composite anode.

“It indicates that the incorporation of Si slightly increases the ID/IG ratio, suggesting a minor in-crease in structural disorder. This can be attributed to the interaction between Si and the carbon matrix, which may induce defects and disrupt graphitic domains. However, this controlled level of disorder is beneficial, as it can enhance lithium-ion storage by increasing active sites while maintaining sufficient graphitic regions for electronic conductivity.”

Figure R1: Raman spectra of HC, HC@Si and HC@Si-P.

We hoped that the reviewer would be satisfied with our responses for publication.

Reviewer 3 Report

Comments and Suggestions for Authors

In this work, phenolic resin was applied as a precursor to synthesis the Si/C composite. A core-shell structure was verified and the final HC@Si-P electrode delivered much improved performance compared to the other groups. Although the method is not novel enough, the systematic study with the multiple characterizations are appreciated to reveal the structure and the mechanism. Since the data support is convincing and logic, with impressive performance improvement after optimization, the reviewer recommends minor review to address the following concerns:

  1. The introduction part is kind of long, especially the second paragraph, it is recommended to divide it logically. Also, the description between Ref 42-44 is too long, please abbreviate.
  2. Please provide the reference of the XRD patterns.
  3. In the Raman test, it showed that while compositing with Si or Si-P, the defect ratio was higher than the control group with resin only, please explain.
  4. There are some writing issues, e.g., in Fig. 4c, there is O-O, Table S2, there is 68/%, please double check your manuscript thoroughly.
  5. Please explain the capacity fluctuation in Fig. 6c.

Author Response

Reviewer: 3

Comments:

In this work, phenolic resin was applied as a precursor to synthesis the Si/C composite. A core-shell structure was verified and the final HC@Si-P electrode delivered much improved performance compared to the other groups. Although the method is not novel enough, the systematic study with the multiple characterizations are appreciated to reveal the structure and the mechanism. Since the data support is convincing and logic, with impressive performance improvement after optimization, the reviewer recommends minor review to address the following concerns:

Response:

Thanks to the reviewer for the general comment on the manuscript and appreciation of the systematic study with multiple characterizations of HC@Si-P electrode material. Based on the suggestion, we have improved the discussion carefully concerning the comments obtained.

  1. The introduction part is kind of long, especially the second paragraph, it is recommended to divide it logically. Also, the description between Ref 42-44 is too long, please abbreviate.

Response:

Thank you for the reviewer’s comments. Based on this feedback, we have carefully revised the introduction by logically dividing the second paragraph for better readability. Additionally, we have abbreviated the description between references [42 to 44] to maintain clarity while preserving the essential information.

“In this vein, Hu. et al. utilized a surface oxidation strategy to improve the electrochemical performance of Si nanoparticles by ……………………………………. potentially improved performance characteristics for battery applications [45]”

We hoped that the reviewer would be satisfied with our responses for publication.

  1. Please provide the reference of the XRD patterns.

Response:

We have now included the appropriate reference for the XRD patterns in the revised manuscript. Thank you for your valuable suggestion, which has helped improve the completeness of our work.

  1. Chen, J; Wang, X; Deng, Z; Kim, E-M; Jeong, S-M. Facile synthesis of Si/C composites for high-performance lithium-ion battery anodes. Nanoscale, 2025,17, 6049-6057.

We have included the above reference in the manuscript appropriately. We hoped that the reviewer would be satisfied with our responses for publication.

  1. In the Raman test, it showed that while compositing with Si or Si-P, the defect ratio was higher than the control group with resin only, please explain.

Response:

Thank you for the reviewer’s comments. The Raman analysis shows that the ID/IG ratio increases upon compositing with Si or Si-P, indicating a higher degree of structural disorder than the hard carbon (HC) derived from resin alone. This increase in defect ratio can be attributed to the interaction between silicon and the carbon matrix, which induces additional defects and disrupts the graphitic domains. Introducing Si and the subsequent pitch-coating process may lead to slight distortions in the carbon structure, increasing the presence of amorphous or defective regions. However, this controlled level of disorder can be beneficial for lithium-ion storage, as it enhances the number of active sites for Li⁺ insertion while maintaining sufficient graphitic regions for electronic conductivity. This balance contributes to the improved electrochemical performance observed in the HC@Si-P composite anode.

We hoped that the reviewer would be satisfied with our responses for publication.

  1. There are some writing issues, e.g., in Fig. 4c, there is O-O, Table S2, there is 68/%, please double check your manuscript thoroughly.

Response:

Thank you for the reviewer’s comments. We have thoroughly reviewed the manuscript and corrected the identified writing issues, including the "O-O" in Fig. 4c and the "68/%" in Table S2. Additionally, we have comprehensively proofread the entire manuscript to ensure clarity and accuracy.

We hoped that the reviewer would be satisfied with our responses for publication.

  1. Please explain the capacity fluctuation in Fig. 6c.

Response:

Thank you for the reviewer’s comments. The capacity fluctuation observed in Fig. 6c (green) can be attributed to the structural and interfacial changes occurring during prolonged cycling. The presence of silicon in the composite anode introduces volume expansion and contraction effects, which can lead to dynamic changes in the solid-electrolyte interphase (SEI) and contact resistance over cycles. Additionally, minor variations in lithium-ion accessibility to active sites and gradual stabilization of the electrode structure may contribute to these fluctuations. Despite these fluctuations, the overall capacity retention remains stable over long-term cycling, indicating that the composite anode effectively accommodates silicon's expansion while maintaining structural integrity. We have now included a discussion on this aspect in the revised manuscript to provide further clarity.

“And the minimal capacity fluctuation observed in Fig. 6c (green) could be attributed to the structural and interfacial changes occurring during prolonged cycling. Additionally, minor vari-ations in lithium-ion accessibility to active sites, as well as gradual stabilization of the electrode structure, may contribute to these fluctuations.”

This discussion can be found on page 11. We hoped that the reviewer would be satisfied with our responses for publication.
